# Parallel Multiple Methods with Adaptative Decision Making for Gravity-Aided Navigation

Shuaipeng Gao  and Tijing Cai *

School of Instrument Science and Engineering, Southeast University, No. 2 Sipailou, Nanjing 210096, China; gsp0803@seu.edu.cn
* Correspondence: caitij@seu.edu.cn

**Abstract:** Gravity-aided navigation is an effective navigation method for underwater vehicles. However, the distribution of the gravity field may affect the measurement errors of gravity anomalies and the precision of gravity-aided navigation. In this paper, the upper and lower thresholds of the gravity field standard deviation are computed by the statistical properties of the local gravity field to classify each grid in the gravity map into different levels. A parallel multiple methods with adaptive decision making (PMMADM) for gravity-aided navigation is proposed which incorporates the gravity anomaly measurements, particle filtering, and maximum correlation method into the observation equation of the extended Kalman filter. The algorithm autonomously selects the observation variables in the filter by combining the gravity field standard deviation at the current position of the carrier with a decision tree. This approach can combine the characteristics of different gravity matching algorithms, reduce the impact of random noise in the measurements, and improve the positioning accuracy of gravity-aided navigation. Physical simulation experiments demonstrated that the proposed gravity matching algorithm achieves the high navigation accuracy and long-term stability in different gravity fields, and the mean value of positioning error is 620.72 m.

**Keywords:** integrated navigation; gravity matching; gravity field standard deviation; skewed distribution; decision tree

## 1. Introduction

Gravity-aided navigation, which utilizes geophysical field information, has attracted significant attention from scholars both domestically and abroad due to its high autonomy and passive concealment features. Key technologies of this navigation system include the pre-production of large-scale high-precision gravity maps, real-time gravity anomaly measurement, and gravity matching algorithms. The basic principle involves utilizing real-time gravity anomaly measurements to locate the carrier position in a known gravity map through the gravity matching algorithm and correcting the position of the inertial navigation system (INS). This approach is applicable to submarine navigation where external radiation information is not available, and can effectively reduce the accumulated error of the INS, ensuring long-term navigation accuracy of the carrier [1].

The performance of gravity-aided navigation is influenced by the characteristics of the gravity field distribution, and for gravity matching algorithms to achieve precise positioning, they require regions with evident gravity field features [2]. Scholars have proposed several evaluation criteria for gravity field regions based on the gravity field's characteristic parameters [3]. Simulation experiments have demonstrated that conducting gravity navigation within adaptive regions significantly improves navigation accuracy [4]. Cai [5] employed the Analytic Hierarchy Process to combine multiple features and obtain gravity field adaptation criteria. In [6], a method was proposed to calculate local gravity features using a moving window and introduce a fast Euclidean distance field algorithm to generate locally adaptive regions. Additionally, Ma [7] introduced an adaptive region selection method based on feature parameter

information entropy. These studies have provided methods for partitioning the gravity field to identify regions with prominent features suitable for gravity matching. Even within the adaptive regions, different gravity matching algorithms exhibit distinct positioning effects. Therefore, depending on the characteristics of the matching algorithms, appropriate algorithms can be selected for navigation in regions with different gravity field characteristics [8].

Currently, there are two main categories of gravity navigation algorithms based on their characteristics: sequential-based matching algorithms, such as Terrain Contour Matching (TERCOM) and Iterative Closest Contour Point (ICCP) [9,10], and iterative-based filter-recursive algorithms, such as Sandia Inertial Terrain-Aided Navigation (SITAN) [11] and Particle Filter (PF) [12]. In pursuit of better applications in gravity-aided navigation, numerous scholars have made improvements to existing algorithms or proposed novel ones by integrating artificial intelligence [13–15]. Zhao [16] considered both gravity measurements and their variation characteristics, enhancing the acquisition method of trajectory points, and optimizing the accuracy and robustness of the PF algorithm. Ouyang [17] analyzed various factors influencing gravity matching navigation results, combining PSO and PF to mitigate the impact of initial registration errors on subsequent particle filtering, thereby enhancing particle filtering navigation accuracy. Mao [18] utilized INS latitude information to decompose gravity anomalies, proposing the V-ICCP and V-TERCOM methods, which averaged a 10% increase in the matching efficiency of the ICCP algorithm. Wang [19] addressed the model error in the state equation of the SITAN algorithm by proposing an adaptive parallel extended Kalman filter-based SITAN algorithm, suppressing filter divergence through adaptive factor-adjusted weights for state prediction information. Different gravity field characteristics and gravity measurement noise can affect the positioning performance of matching algorithms, necessitating the development of noise-resistant and stable gravity matching algorithms [20].

However, these individual methods have certain limitations, and therefore many scholars have proposed ways to combine different types of methods for gravity navigation. For example, Luo [21] proposed a combined the Extended Kalman Filter (EKF) and the PF that completes the gravity matching process in the first layer of filtering and ensures real-time performance in the second layer. Han [22,23] combined the ICCP with the PF and performed an iterative isochronous transformation based on the PF matching results through a two-step calculation. Wei [24] added a weight-based iterative technique to the SITAN using the principles of the TERCOM to reduce the coarse errors in the results of the SITAN. These studies demonstrate that the joint use of multiple methods for gravity navigation can improve the noise immunity and stability of the gravity matching algorithm, resulting in higher positioning accuracy.

Therefore, this paper proposes a parallel multiple methods with adaptive decision making for gravity-aided navigation. Before navigation, the local standard deviation of the gravity map used for navigation is computed to derive a threshold value for level classification of each gravity field region. During the voyage, the algorithm calculates the gravity field standard deviation of the grid in which the carrier is located, and determines the current gravity field level based on the threshold. The algorithm then uses the EKF as the main component in conjunction with the decision tree established in this paper and adaptively incorporates the matching results of the PF and the maximum correlation method (MCM) into the EKF observation equation.

The structure of the paper is as follows. Section 2 is divided into two parts, with the first part providing an overview of the EKF, the PF, and the MCM employed in this study. The second part presents the adaptive selection method for calculating the gravity field level classification threshold and the matching algorithm proposed in this paper. In Section 3, the first part shows the calculation of the threshold parameters for each region in the gravity map using the proposed threshold selection method. The second part presents the results of comparative experiments to validate the improvements in navigation accuracy and long-term stability of the proposed algorithm over traditional methods. Finally, the paper concludes with a summary of the findings and suggestions for further optimization of the proposed method and future research directions.

## 2. Methods

### 2.1. Principle of Gravity-Aided Navigation

The positioning errors of INS will be accumulated, necessitating periodic recalibration or correction using external information to enhance positioning accuracy. Utilizing Earth's gravity field for aiding inertial navigation is a highly promising approach. Gravity-aided navigation involves comparing measured gravity anomaly values during the submarine's movement with stored gravity anomaly data in a computer, leading to more precise determination of the submarine's position. This gravity-aided navigation technique possesses passive, all-weather, and covert characteristics, rendering it well-suited for correcting inertial navigation positions in the absence of external signals, its principle is illustrated in Figure 1.

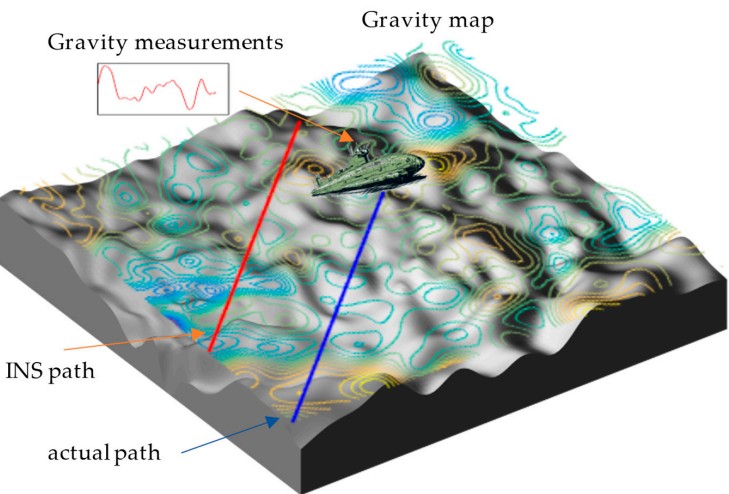

**Figure 1.** Schematic diagram of gravity-aided navigation principle.

The composition diagram of the gravity-aided navigation system is illustrated in Figure 2, comprising the INS, gravimeter, gravity anomaly map, and gravity matching algorithm. The precision of gravity-aided navigation is influenced by factors such as gravity anomaly measurement noise, INS accuracy, gravity map resolution, and precision. As gravity-aided navigation systems gradually transition from theory to practical application, it necessitates gravity matching algorithms to possess high accuracy and real-time capability, enabling long-term stable navigation in a broader range of regions.

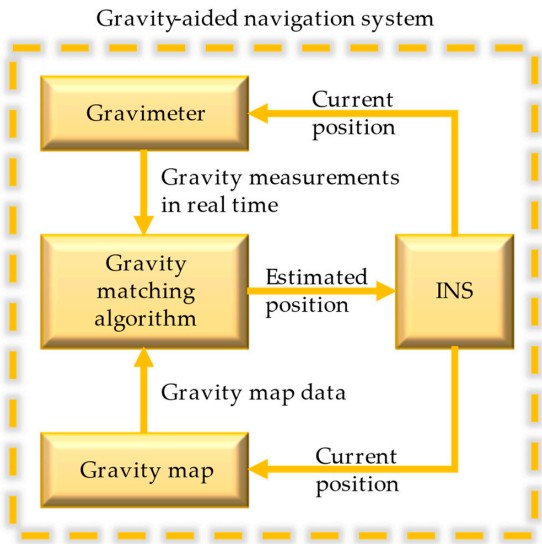

**Figure 2.** Block diagram of gravity-aided navigation system.

### 2.2. Overview of Parallel Methods

The proposed navigation algorithm PMMADM is implemented in an integrated navigation system, which includes an INS, a Doppler velocity log (DVL), and a gravimeter. The algorithm utilizes the EKF with the PF and the MCM for parallel matching computations. The joint gravity matching method is autonomously selected from the three results based on the standard deviation of the local gravity field and the current navigation state of the carrier.

### 2.2.1. Introduction of the EKF Method

The system state variables are chosen as $X = [\delta V_e, \delta V_n, \delta \varphi, \delta \lambda]^T$, where $\delta V_e$ and $\delta V_n$ represent the eastward and northward velocity errors of the carrier, and $\delta \lambda$ and $\delta \varphi$ represent the longitude and latitude errors of the carrier, respectively. The system state equation is as follow:

$$\dot{X} = FX + W \tag{1}$$

where $W$ denotes the system state noise and $F$ means the system state transfer matrix as:

$$F = \begin{bmatrix} \dfrac{V_n}{R_N}\tan\varphi & 2\omega_{ie}\sin\varphi + \dfrac{V_E}{R_E}\tan\varphi & 2\omega_{ie}\cos\varphi v_n + \dfrac{V_e V_n}{R_E}\sec^2\varphi & 0 \\ -2\omega_{ie}\sin\varphi - \dfrac{2V_e\tan\varphi}{R_E} & 0 & -2\omega_{ie}\cos\varphi V_e - \dfrac{V_e^2\sec^2\varphi}{R_E} & 0 \\ 0 & \dfrac{1}{R_N} & 0 & 0 \\ \dfrac{\sec\varphi}{R_E} & 0 & \dfrac{V_e\sec\varphi\tan\varphi}{R_E} & 0 \end{bmatrix} \tag{2}$$

In which $\omega_{ie}$ means the angular speed of the earth's rotation, $R_N$ and $R_E$ are the radius of the curvature of the Uranus and Meridian circles at the position of the carrier.

The measurement vector of the system is:

$$Z = \begin{bmatrix} V_e - V_e^{DVL} \\ V_n - V_n^{DVL} \\ \lambda - \lambda^G \\ \varphi - \varphi^G \\ g(\lambda, \varphi) - \tilde{g} \end{bmatrix} \tag{3}$$

The $V_e^{DVL}$ and $V_n^{DVL}$ represent the east and north velocities of the carrier measured by the DVL, respectively. Meanwhile, $\lambda^G$ and $\varphi^G$ denote the longitude and latitude of the matching results obtained by the PF or the MCM, and $\tilde{g}$ represents the gravimeter measurements. The PMMADM adaptively determines whether the matching result variables and gravimeter measurements in the measurement vector are valid. If only the matching result variable is valid, the EKF method is labeled as EKF-M, with the source of the matching result marked with a subscript. If only the gravimeter measurement is valid, the EKF is labeled as EKF-g. If both are valid, it is labeled as EKF-Mg.

The observation equation for the system is:

$$Z = HX + V \tag{4}$$

where $V$ denotes the system measurement noise and $H$ means the system observation matrix as:

$$H = \begin{bmatrix} I_{2\times2} & 0_{2\times2} \\ 0_{2\times2} & H_G \\ 0_{1\times2} & H_g \end{bmatrix} \tag{5}$$

If the gravity measurements are available, the observation matrix $H_g$ is defined as $H_g = \left[ \frac{\partial g}{\partial \varphi}, \frac{\partial g}{\partial \lambda} \right]$, where $\frac{\partial g}{\partial \varphi}$ and $\frac{\partial g}{\partial \lambda}$ denote the first-order partial derivatives of the gravity field with respect to latitude and longitude, respectively, at the position indicated by the INS. Otherwise, $H_g$ is set to be a zero matrix. If the matching result is available, $H_G$ is set to the unit matrix. By adapting the observation matrix of the EKF, the proposed algorithm utilizes both the gravity anomaly measurements and the gravity matching results from other algorithms to calculate the correction for the INS position.

### 2.2.2. Introduction of the PF Method

The nonlinear model of the PF system is:

$$
\begin{aligned}
x_{t+1} &= x_t + u_t + v_t \\
y_t &= g(x_t) + e_t
\end{aligned}
\tag{6}
$$

where $x_t = [\lambda_t, \varphi_t]^T$ represents the current position of the particle, and $u_t$ denotes the system state transfer variable. The observed variable, $y_t$, corresponds to the gravity anomaly measurement, while $g(x_t)$ represents the gravity map value at the actual location of the carrier. Furthermore, $e_t$ and $v_t$ are independent white noise with known probability density distribution functions $p_{e_t}(\cdot)$ and $p_{v_t}(\cdot)$, respectively.

Suppose that there are $K$ particles $\{x_t(k), k = 1, 2, \ldots, K\}$ distributed in the vicinity of the carrier's location, and the posterior probability density of each particle is represented by:

$$
p(x_t(k)|\mathbb{Y}_t) \quad k = 1, 2, \ldots, K
\tag{7}
$$

where $\mathbb{Y}_t = \left\{ \tilde{g}_i \right\}_{i=0}^t$ denotes the accumulated gravity anomaly measurement obtained by the gravimeter from the start of the navigation to the current moment.

The systematic process noise probability density function for each particle is obtained from carrier displacement as:

$$
\begin{aligned}
p_{v_t}\left( \mu_t^{kj}, u_t \right) &= \frac{1}{\sigma_Q \sqrt{2\pi}} \exp\left( -\frac{\left\| \mu_t^{kj} - u_t \right\|}{2\sigma_Q^2} \right) \\
\mu_t^{kj} &= x_t(k) - x_{t-1}(j)
\end{aligned}
\tag{8}
$$

where $\sigma_Q$ is the system process noise standard deviation related to the INS and the DVL accuracy.

The probability density function of the gravity anomaly measurement noise for each particle is:

$$
p_{e_t}(\Delta g(k)) = \frac{1}{\sigma_R \sqrt{2\pi}} \exp\left( -\frac{\left[ \tilde{g}_t - g(x_t(k)) \right]^2}{2\sigma_R^2} \right)
\tag{9}
$$

where $\sigma_R$ is the gravity anomaly measurement error standard deviation.

Using Bayesian filtering method, the recursive equations for the states and probability density of the particle population are obtained as follows:

$$
x_{t+1}(k) = x_t(k) + u_t
\tag{10}
$$

$$
\begin{aligned}
p(x_t(k)|\mathbb{Y}_t) &= \alpha_t^{-1} p_{e_t}(\Delta g(k)) p(x_t(k)|\mathbb{Y}_{t-1}) \\
\alpha_t^{-1} &= \sum_{k=1}^{K} p_{e_t}(\Delta g(k)) p(x_t(k)|\mathbb{Y}_{t-1}) \\
p(x(k)|\mathbb{Y}_{t-1}) &= \sum_{j=1}^{K} p_{v_t}\left( \mu_t^{kj}, u_t \right) p(x_{t-1}(j)|\mathbb{Y}_{t-1})
\end{aligned}
\tag{11}
$$



Based on the distribution and probability density of the particle population at current moment, the position of the carrier can be estimated as:

$$\hat{x}_t = \sum_{k=1}^{K} x_t(k) p(x_t(k)|\mathbb{Y}_t) \tag{12}$$

The variance of the estimated position is:

$$P_t = \sum_{k=1}^{K} \|x_t(k) - \hat{x}_t\| p(x_t(k)|\mathbb{Y}_t) \tag{13}$$

### 2.2.3. Introduction of the MCM Method

After collecting the navigational position and gravity anomaly measurements at each distance, the carrier obtains a reference track $L_{INS}$ composed of $N$ track points. To search for the matching track, the indicator track is rotated and translated within the specified search area, according to the following formula:

$$L_{\theta d} = C_\theta L_{INS} + \Delta D_d \tag{14}$$

where $C_\theta$ represents the rotation matrix and $\Delta D_d$ represents the offset distance. By varying the rotation angles $\theta$ and offset distances $d$, multiple trajectories $L_{\theta d}$ can be generated for matching within the search range.

The gravity anomaly map values for each track point in $L_{\theta d}$ are obtained using the nearest grid point method.

$$g_M\left(\lambda_{\theta d}^i, \varphi_{\theta d}^i\right) = g(<\lambda_{\theta d}^i>, <\varphi_{\theta d}^i>), \; (\lambda_{\theta d}^i, \varphi_{\theta d}^i) \in L_{\theta d}, i < N \tag{15}$$

where $<\cdot>$ indicates a rounding operation.

This paper uses MSD to evaluate the correlation of gravity anomaly measurement sequences with the trajectory sequences to be matched:

$$J_{MSD} = \sum_{i=1}^{N} (\widetilde{g}^i - g_M^i)^2 \tag{16}$$

Within the specified search range, the correlation between all $L_{\theta d}$ and the gravity anomaly measurement sequences is calculated iteratively. Then, the end point $\hat{x}_N = \left(\lambda_{\theta d}^N, \varphi_{\theta d}^N\right)$ of $L_{\theta d}$ that corresponds to the smallest $J_{MSD}$ is selected as the matching result of the correlation extremum method.

The matching results obtained by the MCM are related to the randomness of the gravity field. Therefore, the end points $x_N$ corresponding to the $n$ smallest correlation values are selected, and their dispersion in distribution is used to estimate the variance of the matching results.

$$P_n = Var\left(\left\{x_N \middle| x_N \Longleftarrow \min_n J_{MSD}\right\}\right) \tag{17}$$

### 2.3. Description of The PMMADM

The algorithm proposed in this paper consists of two main parts: the calculation of gravity field standard deviation thresholds before the voyage, and the calculation of gravity matching during the voyage.

### 2.3.1. Classification of Gravity Field

The accuracy of gravity-aided navigation will be influenced by the distribution characteristics of the gravity field in the navigation area. The main indicators used to evaluate

the gravity field include the standard deviation, slope, and roughness of the gravity field, which reflect its inherent properties from different perspectives. According to [25,26], gravity matching algorithms tend to achieve better navigation accuracy in regions where the gravity field is well-adapted.

In this paper, the standard deviation of the gravity field $\sigma$ is selected as an evaluation index for selecting matching methods in different gravity fields. The standard deviation of the gravity field is a parameter that reflects the intensity of gravity field variation in a local area. The larger the value of $\sigma$, the more intense the gravity field variation is. Assuming that the entire ocean gravity field is gridded, the standard deviation of the gravity field can be calculated as follows:

$$
\begin{aligned}
\sigma &= \sqrt{\frac{1}{mn-1}\sum_{i=1}^{m}\sum_{j=1}^{n}[g(i,j)-\overline{g}]^2} \\
\overline{g} &= \frac{1}{mn}\sum_{i=1}^{m}\sum_{j=1}^{n}g(i,j)
\end{aligned}
\tag{18}
$$

where $m$ and $n$ denote the number of horizontal grid points and the number of vertical grid points in the area of standard deviation calculation, respectively. $g(i,j)$ indicates the gravity anomaly map value at coordinate $(i,j)$.

The above equation is used to calculate the standard deviation of the gravity field for all grid points in a region of the Yellow Sea in China, and the statistical histogram is obtained as shown in Figure 3.

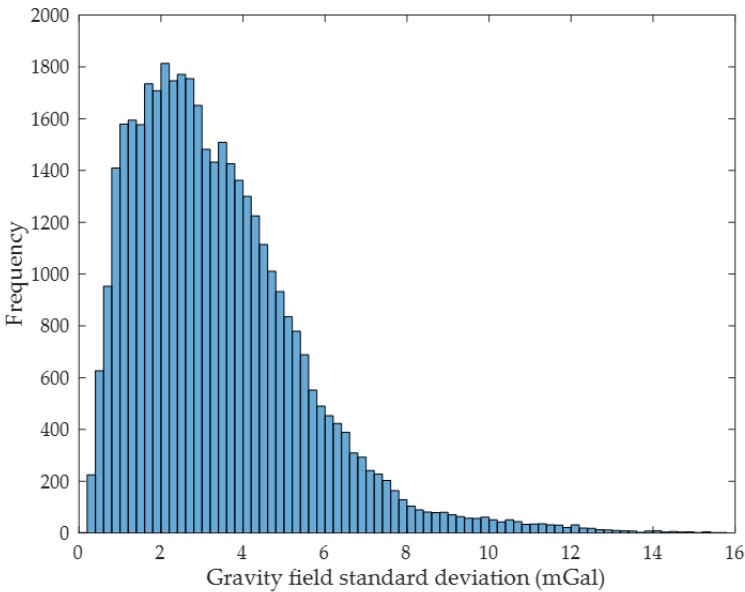

**Figure 3.** Statistical histogram of the standard deviation of the gravity field within a sea area.

As can be seen in Figure 3, the statistical distribution of the standard deviation of the gravity field follows a skewed distribution [27,28]:

$$
f(x;\alpha,\mu,s) = \frac{2}{\sigma\sqrt{2\pi}}\exp\left(-\frac{(x-\mu)^2}{2s^2}\right)\Phi\left(\frac{\alpha(x-\mu)}{s}\right)
\tag{19}
$$

where $s$ and $\mu$ represent the statistical standard deviation and mean of the standard deviation of the gravity field, respectively, $\alpha$ is the skewness parameter, and $\Phi$ is the cumulative distribution function of the standard normal distribution.

$$
\Phi(z) = \frac{1}{\sqrt{2\pi}}\int_{-\infty}^{z}e^{-\frac{x^2}{2}}dx
\tag{20}
$$

The skewness parameter $\alpha$ is estimated using the skewness of the overall sample.

$$\alpha = \frac{\mu_3}{s^3} \tag{21}$$

where $\mu_3$ is the third central moment of the sample data.

$$\mu_3 = \frac{1}{n}\sum_{i=1}^{n}(x_i - \overline{x})^3 \tag{22}$$

where $n$ means the number of grid points. Equation (19) can be used to describe the gravity field standard deviation distribution. In this paper, the upper threshold $U$ and lower threshold $L$ of the gravity field standard deviation are obtained based on its distribution. Consequently, the gravity field in the navigation area is divided into three classes, as shown in Table 1.

**Table 1.** Classification of gravity field levels.

| Level | Description | Condition |
|:---:|:---|:---:|
| I | The gravity field is flat and the features are not obvious. | $\sigma < L$ |
| II | The gravity field varies significantly, and the gravity anomaly measurement noise is small. | $L < \sigma < U$ |
| III | The gravity field fluctuates drastically, but the gravity measurement noise is large. | $\sigma > U$ |

Assuming that the grid share of the gravity field of class II is $\beta$, the lower threshold $L$ and the upper threshold $U$ of the standard deviation for dividing the gravity field classes are obtained from the t-distribution table, respectively [29,30]:

$$\begin{aligned} L &= \overline{x} - \frac{t_{\frac{\beta}{2},n-1}s}{\sqrt{n}}\sqrt{1 + \frac{1}{n}\left(1 - \frac{3}{\alpha^2}\right)} \\ U &= \overline{x} + \frac{t_{\beta/2,n-1}s}{\sqrt{n}}\sqrt{1 + \frac{1}{n}\left(1 - \frac{3}{\alpha^2}\right)} \end{aligned} \tag{23}$$

Before navigation, the gravity map used for gravity-aided navigation is partitioned into regions, and the upper and lower thresholds are computed individually for each region. These thresholds are utilized to categorize the gravity field grids within the region into various classes.

### 2.3.2. The PMMADM Method

The fundamental principles of three gravity-aided navigation algorithms were elucidated in the preceding section. Distinct gravity matching methods exhibit varying performance in gravity fields with diverse characteristics. For instance, the EKF method demonstrates enhanced localization stability in regions characterized by insignificance. Conversely, the PF and the MCM yield superior matching outcomes in regions characterized by substantial fluctuations in the gravity field owing to their robust resistance to noise.

In this study, the PMMADM incorporates the EKF algorithm as the primary component, while simultaneously employing the PF and the MCM methods for gravity matching calculations. The schematic diagram illustrating the composition of the algorithm is presented in Figure 4.

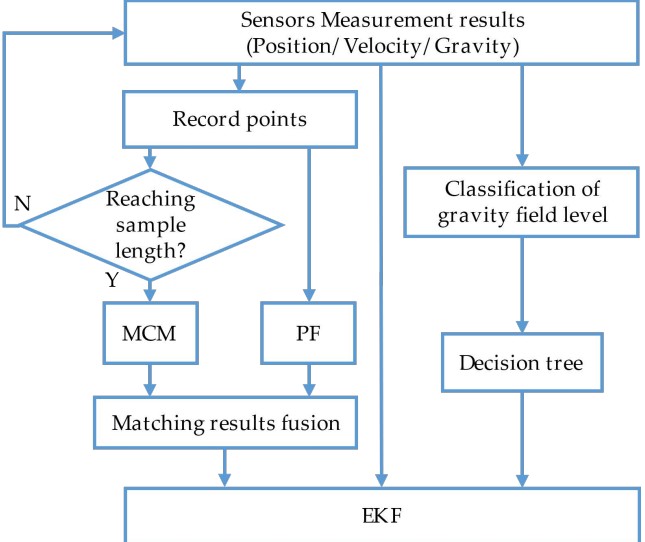

**Figure 4.** Block diagram of the PMMADM.

In the PMMADM, the PF and MCM algorithms record sampling points at regular intervals. Once the number of sampled points reaches the length of the MCM sequence, the matching results, $\hat{x}_t$ from the PF algorithm and $\hat{x}_N$ from the MCM algorithm, are fused to obtain the gravity matching result. The fusion results of the two methods can be obtained from Equations (12), (13), (16), and (17).

$$\hat{\boldsymbol{x}}_M = \hat{\boldsymbol{x}}_t + K(\hat{\boldsymbol{x}}_t - \hat{\boldsymbol{x}}_N)$$
$$K = \frac{P_t}{P_t + P_n} \tag{24}$$

The variance of the fusion results is:

$$P = P_t + K * P_n \tag{25}$$

During the voyage, the PMMADM continuously calculates the standard deviation of the gravity grid at the current location to assess the level of the gravity field. The observation matrix of the EKF algorithm is determined using a decision tree that considers both the navigation state and the gravity field level. Figure 5 illustrates the decision tree specifically designed for the algorithm proposed in this paper. It is used for the adaptive selection of matching results or gravity measurement based on the standard deviation of the gravity field at grid points.

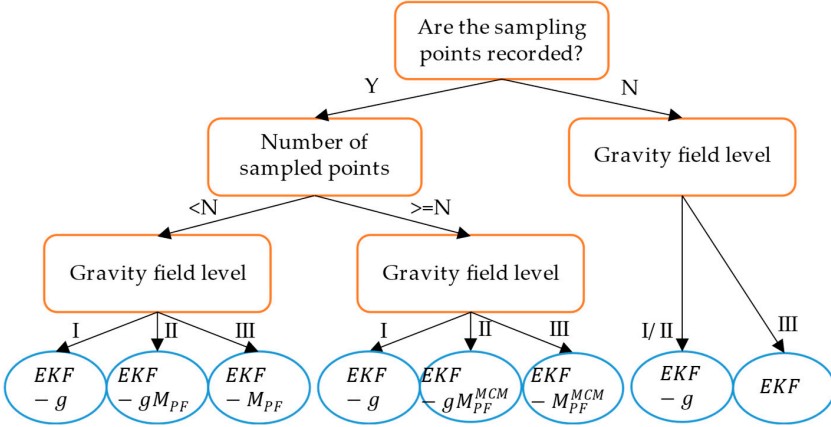

**Figure 5.** Decision tree.

Due to the less prominent characteristics of the Class I gravity field, the utilization of PF and MSD algorithms fails to yield satisfactory matching results. However, in such scenarios, the errors in gravity anomaly measurements and gravity field linearization are small, it is appropriate to employ gravity measurements within the EKF for position correction of the INS. This approach ensures the stability of navigation results by mitigating the influence of the stochastic nature of matching outcomes from other algorithms within the Class I gravity field. Conversely, when the carrier operates within the Class III gravity field, significant errors may arise in gravity anomaly measurements. Consequently, the utilization of gravity measurements as observation variables is discontinued, and Equations (24) and (25) are employed to acquire fused results from various matching algorithms for position correction.

### 3. Experiment and Results

The experimental data used in this study were obtained from actual navigation measurements conducted in a certain sea area of China. The carrier is equipped with a laser INS, DVL, and gravimeter. The stability of the gyroscope zero bias in the INS is $0.003°/h$, with a random walk of $0.0005°/\sqrt{h}$, and the accelerometer bias is less than 5 mg. The position update frequency of INS is 1 Hz. The velocity measurement accuracy of the DVL is $0.4\%v \pm 5$ mm/s and data output frequency of 1 Hz. The gravity meter achieves a continuous measurement accuracy of better than 1.5 mGal at sea, with a gravity anomaly data output frequency of 1 Hz. The GPS receiver records the real-time position of the carrier, which is used to evaluate the navigation accuracy of the gravity matching algorithm. The vessel's average sailing speed is 6.7 m/s, and the duration of the entire experimental data used in the study is approximately 97 h. By employing an ARM-based hardware platform to achieve real-time emulation of sensor data streams, this paper accomplished a semi-physical simulation of gravity-aided navigation within an underwater gravity measurement and processing platform based on OMAP138.

### 3.1. Gravity Field Classification

The gravity map utilized in this experiment is derived from satellite altimetry inversion and exhibits a resolution of $1' \times 1'$. The map encompasses a latitude range of $13.3°$ and a longitude range of $31.3°$, with the maximum gravity anomaly value reaching 451.047 mGal, while the minimum value stands at $-283.4$ mGal. Figure 6 visually presents the gravity map employed in the experiment, accompanied by the GPS trajectory depicting the carrier's navigation path.

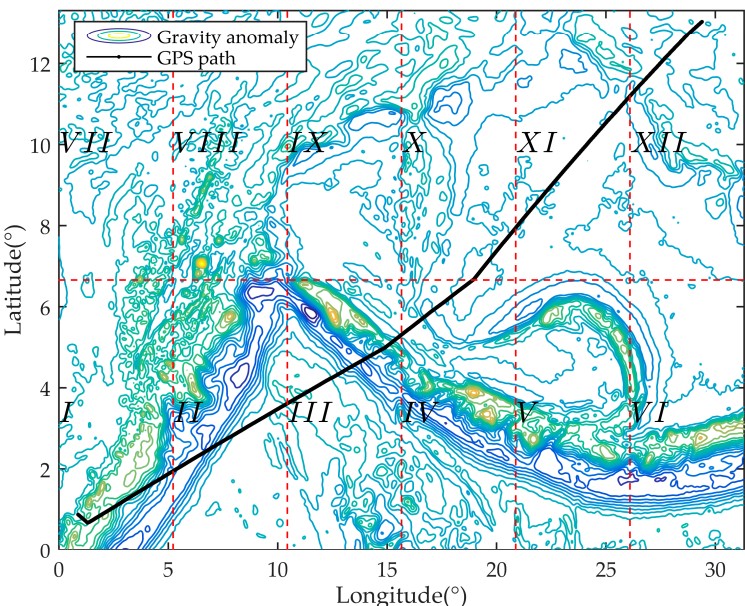

**Figure 6.** Gravity map and GPS path.

In order to minimize computational load, the gravity map is partitioned into twelve regions and labeled sequentially, as depicted in Figure 6. For each region, upper and lower thresholds are separately calculated using the gravity field classification method proposed in this paper. The standard deviation calculation is performed using a $5 \times 5$ grid, with $\beta$ is 90% of the gravity field selected as level II. The computation results for the upper threshold $U$ and the lower threshold $L$ of each gravity field are presented in Table 2.

**Table 2.** Gravity field classification thresholds for each region.

| Serial Number | $L$ (mGal) | $U$ (mGal) |
| :---: | :---: | :---: |
| $I$ | 1.2 | 7.1 |
| $II$ | 0.89 | 9.33 |
| $III$ | 1.48 | 11.49 |
| $IV$ | 1.99 | 11.09 |
| $V$ | 1.48 | 12.3 |
| $VI$ | 0.6 | 10.15 |
| $VII$ | 1.2 | 5.77 |
| $VIII$ | 2.1 | 9.75 |
| $IX$ | 1.2 | 6.33 |
| $X$ | 1.0 | 4.89 |
| $XI$ | 0.7 | 3.13 |
| $XII$ | 0.6 | 3.85 |

From Figure 6, it is evident that the trajectory traverses through seven regions: $I$, $II$, $III$, $IV$, $X$, $XI$, and $XII$. Referring to the data in Table 2, it is noteworthy that the $U$ for the first four regions are considerably higher compared to the other three regions. In regions where the standard deviation of the gravity field falls below the $L$, the gravity field exhibits minor fluctuations and less pronounced characteristics, making it unsuitable for gravity matching calculations using the PF and the MCM methods. Conversely, in regions surpassing the $U$, the gravity anomaly experiences significant fluctuations, and the gravity field characteristics become more prominent. However, there is a higher probability of encountering substantial noise in the gravity anomaly measurements. Therefore, it is not advisable to utilize these measurements for position correction in the EKF algorithm.

The measurement noise of gravity anomalies is related to the standard deviation of the gravity field. In this paper, the classification of gravity fields is employed to identify regions that may exhibit significant measurement errors, allowing for the selection of appropriate gravity matching algorithms. When the carrier is located in grid cells classified as Class III gravity fields, larger measurement errors are prone to occur. Figure 7 presents a comparison between the absolute error of gravity anomaly measurements throughout the carrier's trajectory and the standard deviation of the gravity field in the corresponding regions.

Figure 7 illustrates a clear relationship between the measurement error of gravity anomalies and the standard deviation of the gravity field in the navigation region of the carrier. The threshold lines in each region indicate that significant measurement errors in gravity anomalies primarily occur in Class III gravity fields. In order to improve the accuracy of gravity matching navigation, the proposed method in this paper, which includes gravity field classification and algorithm selection, recommends employing the PF and MCM algorithms in regions characterized by Class III gravity fields. These algorithms demonstrate robust resistance to noise. Conversely, when the standard deviation of the grid where the carrier is located falls below the lower threshold limit of the interval, the noise in gravity anomaly measurements is reduced, making EKF-g the preferred choice for ensuring navigation accuracy and stability.

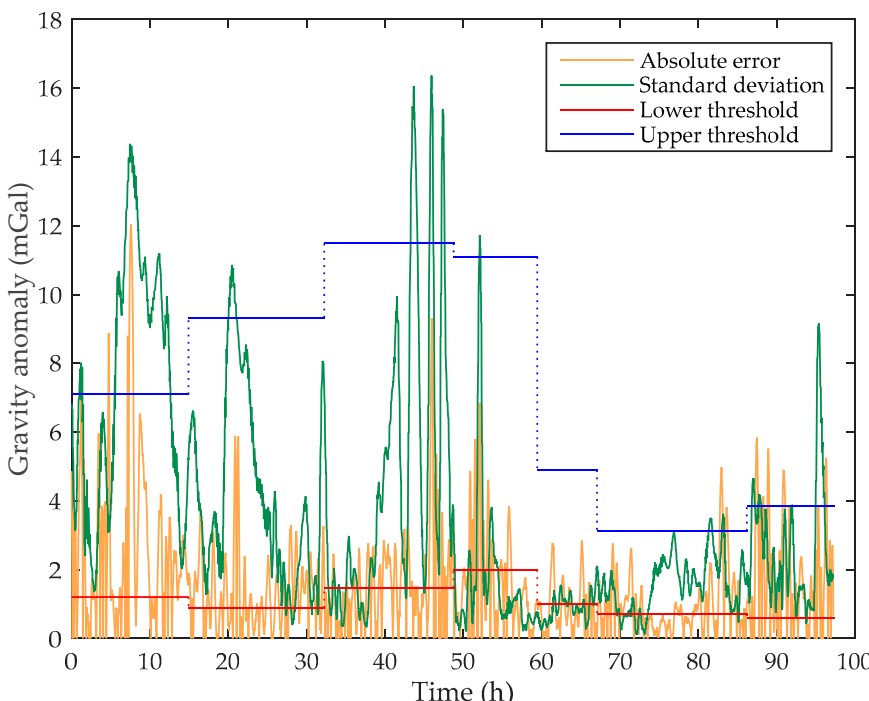

**Figure 7.** Standard deviation of gravity field and absolute error of gravity measurement.

### *3.2. Gravity Matching in Different Gravity Fields*

Within Region *III*, the carrier traversed three distinct levels of gravity field. Based on the advanced threshold division of gravity field in 3.1 and the trajectory of carrier, three segments of the trajectory are selected for comparison. These selected segments were subjected to comparative experiments using the MCM, SITAN, and PF algorithms, as well as the PMMADM algorithm proposed in this paper.

In the SITAN algorithm, the linearization method for the gravity field adopts a nine-point fitting approach, and the initial position is set to the INS indication position. In the PF algorithm, the initial distribution state of the particle swarm follows a normal distribution, with the mean value equal to the inertial guide position and a standard deviation of $0.05°$. For the MCM algorithm, the sampling sequence length is set to 20, the search range radius is two grids, and the sampling sequence is updated using a sliding window. In order to maintain consistency in the comparison experiments, the parameter settings for the PMMADM algorithm remain the same as those used for each algorithm in the comparison.

#### 3.2.1. Track Segment I

The standard deviations of the gravity map grid points encountered in this trajectory segment are all below the lower threshold of Region *III*. This indicates a relatively flat gravity field with indistinct features in this region. The latitude and longitude errors obtained by the matching algorithms are shown in Figure 8.

Both the MCM and PF algorithms fail to accurately determine the true position of the carrier and can only perform gravity matching calculations when there are trajectory points, resulting in discontinuous outputs for gravity navigation. On the other hand, the SITAN algorithm exhibits a smooth and continuous error curve, although it also falls short in precisely determining the carrier's actual position. However, it demonstrates favorable real-time performance and stability.

As the gravity field encountered in this segment is classified as Level I based on the decision tree, it indicates that the PMMADM algorithm, in this region, only employs EKF-g for corrections. Consequently, its matching results align with those of the SITAN algorithm and it isn't shown in the Figure 8.

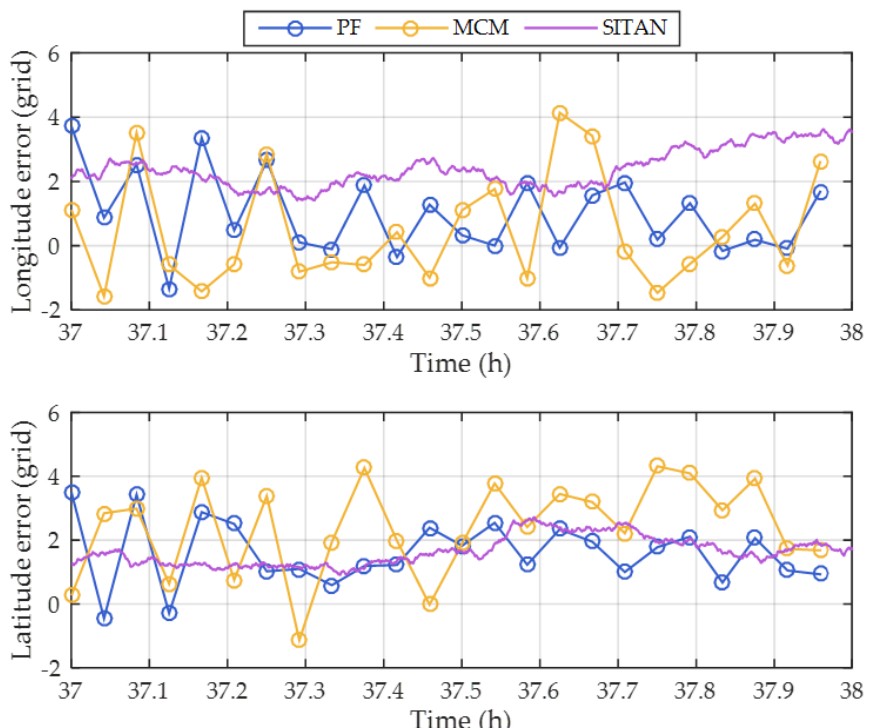

**Figure 8.** Positioning errors for track segment I.

### 3.2.2. Track Segment II

In this segment of the trajectory, the carrier is situated in a region with a relatively large standard deviation of the gravity field, while the maximum value of gravity measurement error does not exceed 3 mGal. At this stage, the carrier is positioned in an area with distinct characteristics and minimal measurement errors, making it well-suited for conducting gravity matching calculations to achieve higher positioning accuracy. The latitude and longitude errors obtained from the four algorithms are illustrated in Figure 9.

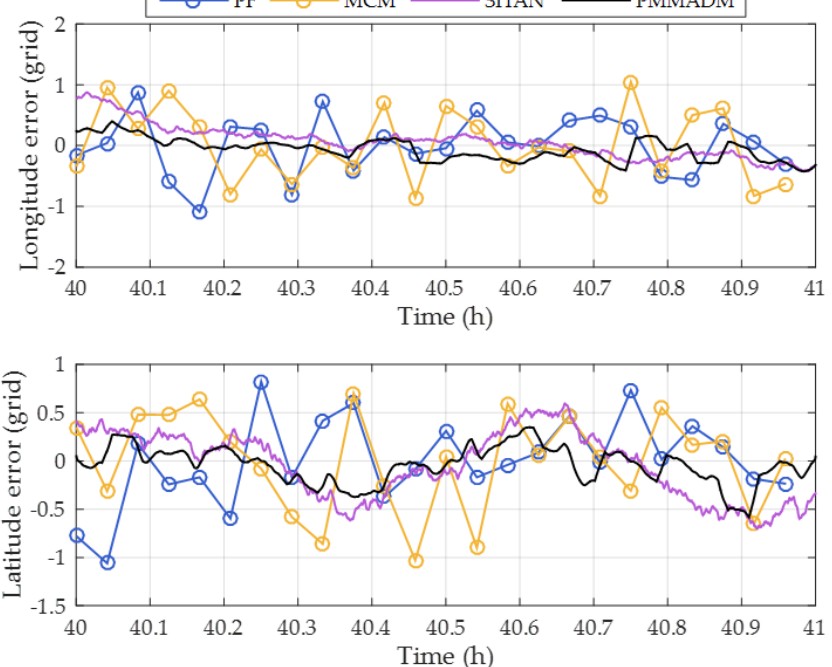

**Figure 9.** Positioning errors for track segment II.

As depicted in Figure 9, when the gravity measurement errors are small and the gravity field exhibits distinct features, all four gravity navigation algorithms confine the positioning errors within a single grid, thereby achieving effective gravity-aided navigation. Table 3 presents the statistics of the eastward and northward distances between the matching positions of the four algorithms and the GPS positions. The data in Table 3 indicate that within the adapted region, the proposed algorithm demonstrates the highest positioning accuracy and stability.

**Table 3.** Segmentation statistics of positioning errors.

| Method | Eastward Error | | Northward Error | |
|---|---|---|---|---|
| | Mean (m) | Std (m) | Mean (m) | Std (m) |
| PF | 713.96 | 540.47 | 635.82 | 528.37 |
| MCM | 967.92 | 570.13 | 766.46 | 541.43 |
| SITAN | 389.84 | 338.02 | 557.41 | 320.19 |
| PMMADM | 281.73 | 206.22 | 272.38 | 229.05 |

### 3.2.3. Track Segment III

During this trajectory segment, the carrier transitions from a region with gravity classified as level II to another region with level III gravity, and there are significant variations in the encountered gravity field. Towards the end of the trajectory, post-data processing reveals large random errors in the gravity anomaly measurements. The matching results obtained from the four different methods are illustrated in Figure 10.

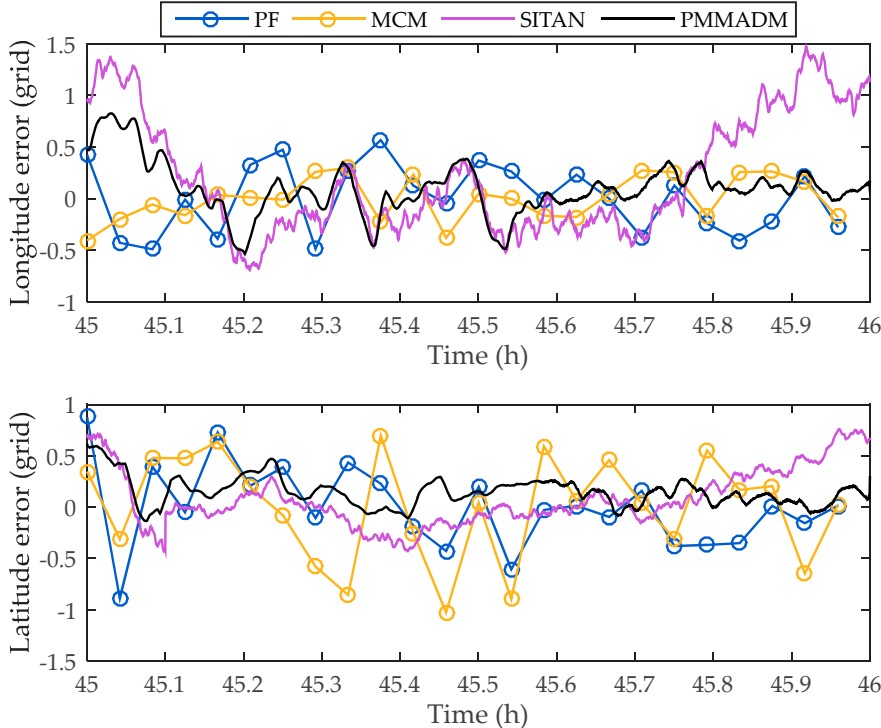

**Figure 10.** Positioning errors for track segment III.

At the beginning of this trajectory segment, the longitude error is within one grid, and the latitude error is 0.7 grid. Within the adaptable region and experiences relatively low gravity measurement noise in the beginning, both SITAN and PMMADM quickly converge to the vicinity of the true position. PF and MCM algorithms also exhibit a similar trend of reducing the INS position error during the initial stage. Towards the end of the trajectory, a sudden increase in measurement noise leads to the divergence of SITAN's longitude

error, and the latitude error gradually increases, no longer showing a trend of convergence towards the true position. From Figure 10, it is evident that PF, MCM, and PMMADM exhibit robustness against noise at this stage. Their positioning errors remain stable, and they can still perform correct matching navigation.

### 3.3. Gravity Matching Navigation under Long Time

To investigate the performance of the proposed algorithm in continuous gravity matching during long-term navigation, this experiment conducted continuous matching navigation using the entire voyage data. In the matching process, to prevent mismatching that could disrupt the normal functioning of traditional algorithms, we introduced DVL velocity as an additional constraint for the three comparative algorithms used in Experiment 3.2.

With these modifications, gravity matching navigation is performed independently using these algorithms, and the resulting navigation outcomes are compared with the GPS positions. The errors in latitude and longitude are depicted in Figures 11 and 12, respectively.

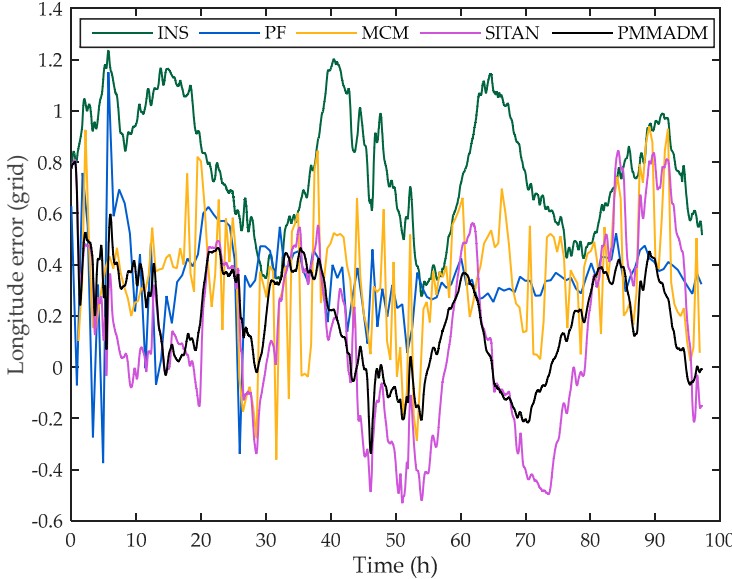

**Figure 11.** Longitude error.

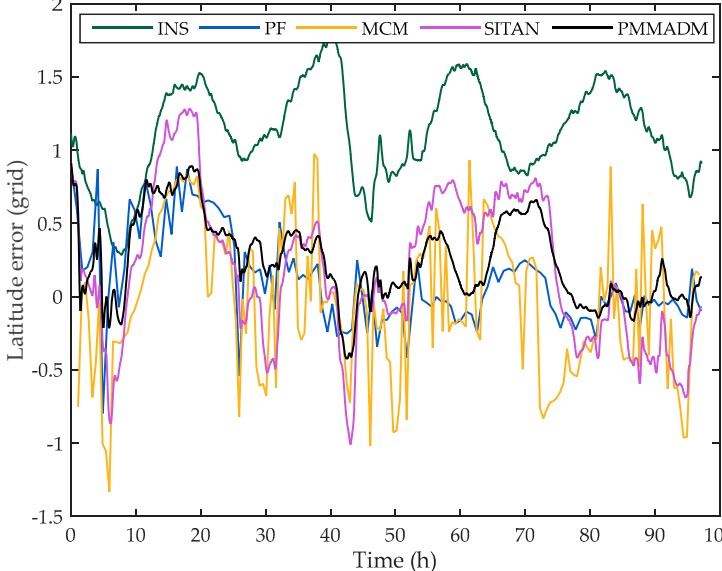

**Figure 12.** Latitude error.

From Figures 11 and 12, it can be observed that the INS exhibits initial position errors, with a longitude error of 0.8 grid and a latitude error of 1 grid. Without external position information for correction, the navigation error curve of the INS fluctuates periodically, rendering it ineffective for accurate navigation positioning. The SITAN algorithm corrects the initial position error early in the voyage. However, due to the influence of gravity anomaly measurement noise, the SITAN algorithm experiences rapid error divergence in certain regions. The PF algorithm demonstrates significant random matching errors in the initial stage, as the particle swarm carries limited genetic information. As the particle swarm gradually evolves, the matching errors in the later stages of the voyage exhibit reduced fluctuations, but a fixed error remains. The error curve of the MCM method fluctuates around zero, but it often produces incorrect matching points, indicating relative instability compared to other algorithms.

All three algorithms have limitations in achieving gravity-aided navigation in certain regions. However, PMMADM overcomes these limitations by employing decision judgment to avoid regions with flat gravity fields or obvious gravity anomaly measurement noise, resulting in more effective gravity matching navigation. The error curves of PMMADM demonstrate rapid convergence to a position error close to zero at the beginning of the navigation. Throughout the entire voyage, the error curves remain smooth and stable. After 10 h, the longitude error does not exceed half a grid, and after 20 h, the latitude error does not exceed 0.7 grid, indicating a high level and stable of positioning accuracy achieved by PMMADM.

Based on the navigation area of the carrier, the trajectory of the route in areas I–IV is classified as the first section, while the trajectory of the route in areas XI–XII is classified as the second section. The first section exhibits distinct gravity field characteristics, whereas the second section experiences a flat gravity field. Table 4 presents a comprehensive statistical analysis of the relative position errors for the two trajectories using different gravity matching methods.

**Table 4.** Segmentation statistics of positioning errors.

| Method | Seg 1 | | Seg 2 | |
|---|---|---|---|---|
| | **Mean (m)** | **Std (m)** | **Mean (m)** | **Std (m)** |
| INS | 2570.03 | 565.43 | 2601.36 | 426.06 |
| PF | 982.16 | 439.72 | 706.90 | 111.46 |
| MCM | 1379.03 | 582.16 | 1739.92 | 788.39 |
| SITAN | 1260.99 | 570.07 | 1720.57 | 747.88 |
| PMMADM | 857.89 | 423.14 | 620.72 | 311.21 |

It is evident from Table 4 that the PMMADM algorithm exhibits superior stability and accuracy in terms of localization error and is less susceptible to the influence of gravity field characteristics, regardless of whether it is in regions with significant gravity field variations or in flat regions. In the first trajectory segment, both the MCM and SITAN algorithms demonstrate smaller average matching errors compared to the latter half of the trajectory. In the second trajectory segment, the particle swarm evolution of the PF algorithm is sufficient, resulting in higher localization accuracy and stability when compared to MCM and SITAN. Remarkably, the PMMADM algorithm consistently achieves comparable levels of localization accuracy across gravity fields with different characteristics, showcasing superior stability and broader applicability.

## 4. Conclusions

This paper introduces a parallel multi-method adaptive decision method for gravity matching navigation. By computing the standard deviation thresholds of the gravity field within the navigation area, the algorithm classifies the gravity field into three levels. Based on the current navigation state and the gravity field level at each grid point, the algorithm autonomously determines the appropriate gravity matching method to be used in the

navigation system. This methodology effectively leverages the unique strengths of different gravity matching algorithms in various gravity field scenarios, expanding the applicability of gravity-aided navigation.

The experiments conducted in this paper demonstrate that the proposed algorithm effectively reduces the impact of high noise levels in gravity field measurements by classifying the gravity field. As a result, the algorithm enhances the robustness of gravity matching navigation in noisy environments. The navigation tests further confirm that the algorithm, integrated into a navigation system comprising INS, DVL, and gravimeter, enables continuous and long-term navigation capabilities.

However, it is important to acknowledge the limitations of the algorithm proposed in this study. Firstly, the precomputation of gravity field standard deviation thresholds may involve redundant calculations. Exploring the possibility of real-time computation of thresholds in the proximity of the inertial navigation system's position could reduce computational burden. Secondly, there is potential for further advancements in the individual branches of the adaptive algorithm, with the goal of improving both localization accuracy and real-time performance of the matching algorithm.

**Author Contributions:** Conceptualization, S.G. and T.C.; methodology, S.G.; software, S.G.; validation, S.G. and T.C.; resources, T.C.; data curation, T.C.; writing—original draft preparation, S.G. and T.C.; writing—review and editing, T.C.; supervision, T.C.; project administration, T.C.; funding acquisition, T.C. All authors have read and agreed to the published version of the manuscript.

**Funding:** This work is supported by the National Key R&D Program of China under Grant Nos. 2017YF0601601.

**Institutional Review Board Statement:** Not applicable.

**Informed Consent Statement:** Not applicable.

**Data Availability Statement:** The data presented in this study are available on request from the corresponding author. The data are not publicly available due to privacy reasons.

**Acknowledgments:** Sincere thanks are expressed to all members in the laboratory for their kind support in a successful experiment.

**Conflicts of Interest:** The authors declare no conflict of interest.

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
