# Peer review of "Parallel Multiple Methods with Adaptative Decision Making for Gravity-Aided Navigation"

_jmse, doi:10.3390/jmse11081624_

Round 1

Reviewer 1 Report

I do not detect drawback in the manuscript, and find it relevant to the scientific community. 

The English should be checked by an expert.

Reviewer 2 Report

Dear Authors:
Thank you for your interesting and well written paper.
It clearly presents a parallel multi-method adaptive decision approach for gravity matching navigation.
The algorithm divides the gravity field into three levels based on standard deviation thresholds,
combining different gravity matching algorithms' strengths in different scenarios.
This effectively reduces the impact of high noise levels in gravity field measurements and enhances the robustness of gravity matching navigation in noisy environments.
The e
xperiments demonstrate the algorithm's effectiveness and its integration into a navigation system with INS, DVL,
and gravimeter enables continuous and long-term navigation capabilities.
the proposed PMMADM algorithm is compared with three other algorithms, namely SITAN, PF, and MCM, and the improvenet of the developed approach is clearly demonstrated..

The experimental analysis, applications, and comparison are well presented and discussed.

The paper also acknowledges two limitations of the algorithm, and suggestes the ways to overcome these limitations in the future.

Only two simple comments have been noticed as follows:
1- the abbreviation PMMADM, is better to be defind at first appearance on page 1, line 12.

2- In Figure 5, Are the upper and lower thresholds correctly specified ?
3- Can you recommend the application of your approach for other types of problems ?

Best Wishes

Reviewer 3 Report

This study presents a study on real-time gravity matching navigation using semi-physical simulation experiments and actual measurement data from a sea area in China. The study incorporated the use of the PMMADM algorithm, which is compared with three other algorithms, namely SITAN, PF, and MCM. While the overall work is interesting and well presented, there are a few areas that need improvement before acceptance of the manuscript. 

Specifically, the paper lacks a thorough discussion of the limitations and potential sources of bias in the study. For instance, the authors mention that the carrier's trajectory traverses through specific gravity regions, but there is no discussion of how the trajectory's selection might have influenced the algorithm's performance.

In addition, the three algorithms selected comparison do not provide sufficient substantiation of the PMMADM algorithm, and other well-established gravity matching algorithms should have been included for a more comprehensive evaluation.

Therefore, I would like the following concerns to be addressed before the paper is considered for publication. 

1. The authors should greatly expand Section 3.2, providing more details regarding the comparison with other algorithms, with additional data validation. Other algorithms should also be considered for comparison, including the CADM algorithm. 

2. Additional explanation should be given for why the other algorithms experienced discrepancies, such as the SITAN algorithm experiencing rapid error divergence in a certain region. Why did this occur? Could the authors provide more explanation?

3. The method description for the real-time gravity matching navigation replicated through semi-physical simulation experiments using actual measurement data obtained from a sea area in China needs to be significantly expanded. From the current description, it is difficult to determine the experimental conditions and equipment used to conduct the semi-physical simulation experiments. Therefore, the methods section needs to be expanded to cover this information, as it is currently unclear how the experiment was conducted. Some information is presented in the results and discussion section, but this needs to be included in the methods, not the results and discussion section, and more detail is needed.

The manuscript would benefit from editing by a native English speaker, due to some inconsistencies in grammar and tense.

Reviewer 4 Report

In this paper, the authors proposed a parallel multiple methods e.g., MCM, PF and EKF with adaptive decision making in gravity aided navigation to accurately localize the position of a ship. The authors showed promising results. However, there are certain points that needs further clarification.

1.     Before navigation, the local standard deviation of the gravity map used for navigation is computed to derive a threshold value N for level classification of each gravity field region. What is the threshold value used in the paper is not clear? Is it heuristically chosen? If yes, what is the reason behind that heuristic value?

2.     Related work of gravity aided localization is insufficient. Only in the Introduction section, some works are mentioned that does not depict the real scenario of the research field. The authors should introduce the related work section and make a clear comparison with their work with other related work. Also, add a table of comparison with other related works.

3.     After the Introduction section and Related work section, the authors can include one section about how gravity aided navigation works with suitable figures.

4.     In the last line of abstract, it is stated that, “Physical simulation experiments demonstrated that the proposed gravity matching algorithm achieves the high navigation accuracy and long-term stability in different gravity fields.” The authors should use specific numerical values of accuracy obtained through simulation instead of writing “high accuracy”.

Minor corrections are needed.

Round 2

Reviewer 4 Report

The authors address all my concerns